# Mammary Cistern Size during the Dry Period in Healthy Dairy Cows: A Preliminary Study for an Ultrasonographic Evaluation

**DOI:** 10.3390/ani10112082

**Published:** 2020-11-10

**Authors:** Francesca Bonelli, Chiara Orsetti, Luca Turini, Valentina Meucci, Alessio Pierattini, Micaela Sgorbini, Simonetta Citi

**Affiliations:** 1Department of Veterinary Sciences, via Livornese snc, San Piero a Grado, 56122 Pisa, Italy; francesca.bonelli@unipi.it (F.B.); luca.turini@phd.unipi.it (L.T.); valentina.meucci@unipi.it (V.M.); alessio.pierattini@gmail.com (A.P.); micaela.sgorbini@unipi.it (M.S.); simonetta.citi@unipi.it (S.C.); 2Centro di Ricerche Agro-Ambientali “E. Avanzi”, University of Pisa, via Vecchia di Marina 6, San Piero a Grado, 56122 Pisa, Italy; 3Istituto Zooprofilattico Sperimentale del Lazio e della Toscana “M. Aleandri”, 00178 Rome, Italy

**Keywords:** udder ultrasound, mammary gland cistern, udder cistern size, dairy cows, dry period

## Abstract

**Simple Summary:**

The dry period is a crucial moment for dairy cows since the udder may develop pathological conditions that could influence the next lactation. We propose a preliminary study for the ultrasonographic evaluation of the udder cistern during the dry period in healthy dairy cows. All four udder cisterns were evaluated and measured by ultrasound at different times. Our results showed a statistically significant negative linear correlation between the time and udder cistern size. In addition, this study showed that the udder cistern size decreased throughout the whole dry period and started to increase at the beginning of the next lactation.

**Abstract:**

We evaluated the udder cistern (UC) size during the dry period using ultrasound. Forty healthy quarters were evaluated in both the longitudinal and cross-section of the UC. Quarters were evaluated at the drying-off (T0) and 24 h later (T1), then regularly until the end of the dry period (T7–T58), during the colostrum production phase (TCPP) and at 7 days in milking (T7PP). The Spearman test was applied to find the correlation between the ultrasonographic UC size (UUCS) assessment and time. The Friedman test and Dunn’s test for multiple comparisons as a post-hoc test were performed to compare the forequarter and hindquarter cross-sections (FQCSs and HQCSs, respectively) and the forequarter and hindquarter longitudinal sections (FQLSs and HQLSs, respectively) at T0 vs. T58 vs. TCPP vs. T7PP. A total of 440 images were evaluated. A negative linear correlation between time and FQCS and FQLS (r = −0.95; *p* < 0.0004) and between time and HQCS and HQLS (r = −0.90; *p* < 0.002) was found. The UUCS decreased throughout the dry period, starting to increase at the beginning of the next lactation. Measuring the UUCS provides useful information for monitoring the dry period.

## 1. Introduction

The dry period is the nonlactating period prior to parturition in dairy cows. This period usually begins around 305 days in milking (DIM) and in cows in Europe and United States of America it typically lasts about 60 days [1,2,3]. During the dry period, the mammary gland undergoes tissue remodeling, which is a critical phase for the next lactation and considerably impacts udder health [4]. During the transition from lactating tissue and the (partially) involuted tissue, the mammary gland tends to have a higher risk of infection [5]. Thus, despite a good dry period management, some animals can develop new intramammary infection and may show mastitis in the next lactation [6,7]. Mastitis leads to huge economic losses, especially due to the decrease in milk production and in milk quality [8,9,10].

The first step for evaluating any disease is a complete physical examination; however, in some cases, further diagnostic evaluation may be needed. Ultrasound is a non-invasive technique that allows a real-time visualization of various tissues [11] and helps in identifying pathologic alterations [12]. Ultrasonography has been used to evaluate various internal structures of the udder, such as the udder cistern (UC) of sheep [13,14,15], cows [16] and goats [17], as well as the teat [10,18,19,20]. An ultrasonographic evaluation of the UC size has been studied in both dairy cows [16,21] and sheep [22,23]; however, no studies have evaluated the ultrasonographic measures and aspects of the UC at different times and during the dry period. This is a preliminary study aimed at evaluating the UC size during the dry period by ultrasonography in healthy cows.

## 2. Materials and Methods

The research protocol was approved by the Institutional Animal Care and Use Committee of the University of Pisa (33479/2016). The study took place at the dairy farm of the University of Pisa (Centro di Ricerche Agro-Ambientali “E. Avanzi”). The owner’s written consent was obtained for all the cows included in this study.

### 2.1. Animals and Management

The study evaluated a total of 40 quarters from 10 Italian Friesian cows. All the cows were subject to the same management conditions. Cows calved all year round. Heifers and first lactation cows were bred with semen from an Italian Friesian bull, while cows after the first lactation were bred with a Limousine bull due to the higher market price of the calf. The rearing heifers were bred on the farm. The voluntary waiting period was 55 days. Cows ready for artificial insemination were inseminated by the referring vet. Cows were housed in a free-stall barn with no cubicles. Cows were grouped as follows: heifers, lactating cows up to 120 DIM, lactating cows more than 120 DIM and dry cows.

The building and cleaning management were the same for all the animals. The feeding area was on a soil surface and was sufficiently large to accommodate 5–10 feeding places. The resting area was made of concrete and covered in straw. The straw was changed twice per week, and between changes, the farmer removed the superficial dirtier part. All the lactating cows received the same total mixed ration twice a day and had free access to fresh water. The dry cows received their own special diet. Two weeks before calving, cows were moved into a calving pen made of concrete and covered in straw. A close-up ration was provided to these cows. Cows were milked in a Herringbone milking parlour twice a day, at a milking interval of almost 11 h (at 5 a.m. and 4 p.m.).

### 2.2. Inclusion Criteria

The animals were enrolled on the basis of the following inclusion criteria: (1) not heifers; (2) abrupt drying-off; (3) not affected by mastitis or other diseases diagnosed at drying-off or during the whole study period (healthy quarter); and (4) having four “working” quarters.

Each cow underwent a complete physical examination almost one hour before the time of drying-off. Drying-off was performed during the milking session. Each udder and quarter were visually examined and palpated. An expert operator (LT) then gave a “teat end condition” (TEC) score for each teat [24], and performed the California Mastitis test (CMT) [25] as well as an individual quarter somatic cell count (SCC). Based on the results of the udder, teat and milk assessments, each healthy quarter received 2.6 g of Bismuth subnitrate (Easiseal, Zoetis Italia s.r.l., Rome, Italy) as a sealant. The quarter was considered healthy if there were no signs of inflammation (redness, swelling and pain), the TEC was less than 1 [24], the CMT was 0 [26] and the single quarter SCC was less than 200,000 cells/mL [27].

### 2.3. Ultrasonographic Examination of the Udder

Ultrasounds were performed in a quiet and dark room, with the cows standing normally. The udders were not clipped or cleaned beforehand. In order to visualize the UC of each quarter, the udder was scanned in B-mode using a portable instrument (Mindray DP30Vet, Shenzhen, China) equipped with a convex probe (3.5–5.0 MHz). Ultrasound gel was used to gain a better contact between the probe and the skin. The probe was placed at the teat–udder insertion in order to visualize the UC of each quarter. Both the longitudinal and cross-section views were evaluated, and the UC area was calculated (cm^2^). First the probe was held parallel to the teat for the longitudinal section, and then a 90° rotation was used for the cross-section view [21]. Each image of the UC was recorded and stored in an external hard disk. All the animals were evaluated at the drying-off (T0) time and 24 h later (T1), then once a week until the end of the dry period (T7, T14, T21, T28, T38, T48, T58) and, finally, once during the colostrum production phase (TCPP) and at 7 days of milking (T7PP).

At the same time as the ultrasound, each quarter was examined in order to check its health status. During the dry period it was obviously not possible to obtain milk samples, thus the quarters were considered healthy if the udder and teat showed no signs of inflammation (swelling, redness and pain). On the other hand, it was possible to perform the CMT, TEC and SCC evaluation at T7PP. 

The ultrasound images were saved on a USB pen drive (SanDisk Cruzer Gilde™ 32 Gb, Milpitas, CA, USA) and transferred to a computer (Acer Aspire 5750G, Taipei, Taiwan) for subsequent evaluation. 

### 2.4. Ultrasonographic Images Evaluation

The pictures were processed using ImageJ (National Institute of Health, Bethesda, MD, USA). For each ultrasound image, the outline of the UC was traced freehand by following its echogenic margin; the margin appeared clearly visible thanks to the anechoic content. The software automatically processed the cropped image determining the area of the UC expressed as cm^2^ (Figure 1). In addition, the content and echogenicity of the surrounding mammary tissue were assessed for each ultrasonographic image.

### 2.5. Statistical Analysis

The distribution of the data was evaluated by the Kolmogorov–Smirnov test and the data were expressed as the median and range or as the mean and standard deviation (SD). The data collected were then analyzed with the Spearman test to check the correlation between the ultrasonographic UC size (UUCS) and time (T0–T58). In order to compare the forequarter and hindquarter cross-sections (FQCSs and HQCSs, respectively) and the forequarter and hindquarter longitudinal sections (FQLSs and HQLSs, respectively) at T0 vs. T58 vs. TCPP vs. T7PP, a Friedman test with a Dunn’s test for multiple comparison were used.

A value of *p* < 0.05 was considered statistically significant. Statistical analyses were carried out using GraphPad Prism 6 (San Diego, CA, USA).

## 3. Results

### 3.1. Animals

All the animals in the study were multiparous Italian Friesian cows with a mean ± SD age of 5.6 ± 1.6 years and two lactations. The average weight was 722 ± 64 kg, and the average BCS was 3.0 ± 0.5.

At the beginning of the research protocol, 15 cows were enrolled; however, five animals were excluded because they showed mastitis in one of more quarter at any moment during the study. The mean ± SD SCC at T0 for the cows included was 170 ± 83 cell/µL, while the mean ± SD SCC at T7PP was 194 ± 98 cell/µL.

### 3.2. Udder Cistern Size

The ultrasound technique used was easy to perform in field conditions. The cows remained quiet and standing during the ultrasound examination. The visualization of the UC was optimal in every case. A total of 440 images were measured. Data concerning the measurements of the UC did not show a normal distribution. The median, maximum and minimum values for the front and rear quarters (left and right) in the longitudinal section are reported in Table 1, while Table 2 shows results concerning the cross-section.

A statistically significant negative correlation was found between the time and ultrasound size of the UC for both FQLS and FQCS (r = −0.95; *p* < 0.0004; Figure 2) and for HQLS and HQCS (r = −0.90; *p* < 0.002; Figure 3).

### 3.3. Statistical Analysis

Friedman’s test was statistically significant for FQCS, HQCS, FQLS and HQLS (*p* < 0.0001). The results concerning Dunn’s multiple comparison test are reported in Figure 4, Figure 5, Figure 6 and Figure 7. A statistically significant difference was found between T0 vs. T58, T58 vs. T7PP and TCPP vs. T7PP, while no differences were found between T0 vs. TCPP for FQCS, HQCS, FQL and HQL, and between T58 vs. TCPP for HQL. The T0 was statistically significantly different from T7PP for FQCS and HQL, while no differences were found for FQL and HQCS.

### 3.4. Evaluation of the Ultrasonography Images

The visual aspect of the forequarters’ UC changed during the dry period, passing from a wide arborized pattern to an elongated/round shape, while the rear quarters showed a less marked decrease in UC arborization. From the 38th day of the dry period, the UC margins passed from being clear to being blurred. At TCPP, the UC margins resumed their initial definition. No images showed an increase in the echogenicity of the UC contents and the surrounding mammary tissue (Figure 8).

## 4. Discussion

Ultrasound allows a real-time visualization of various tissues [11] and has been used to evaluate various internal structures of the udder such as the UC [16] and the teat [10,18,19,20]. Several studies have performed ultrasounds of the UC to demonstrate normal mammary gland features in dairy heifers [11,28] and in lactating cows [16,21,29,30], and other authors have evaluated the non-physiological pattern in a sick mammary gland [31]. To the best of our knowledge, no studies have evaluated the size of the UC in dry cows.

There have been two different approaches for the ultrasound evaluation and measurement of the UC: one required the immersion of the udder in water [16], while in the second one, the probe was held directly in contact with the skin of the udder quarter [21]. In this study, we used the second method in order to avoid the use of water, which can lead to an increased risk of mastitis. In fact, water can carry dirt, leading to bacteria penetration through the teat orifice and channel [16,32]. In agreement with the literature [21], the technique was easy to perform in field conditions throughout the study, and the UC was easy to visualize.

Previous studies have only investigated the UUCS in bovines before and after milking, thus we were only able to compare our results obtained at T0 and T1, which were similar to others [21]. At T1, the UUCS was higher than at any other time, but it can be considered to be physiological due to the accumulation of milk after the abrupt dry-off. The UUCS of both the fore and rear quarters in the longitudinal and cross sections decreased throughout the dry period; the UUCS started to increase again at the beginning of the new lactation (TCPP and T7PP). These data differ from findings reported on the histological changes occurring in the bovine mammary tissue during the dry period [1]. Studies on UC cellular proliferation showed that the UC cellular involution during the dry period peaked at 25 days after the dry-off. Moreover, in pregnant cows, the cellular growth starts again at the beginning of lactation [1,33]. This difference could be due to a discrepancy between the UC cellular proliferation and the ultrasonographic evaluation of the UC.

The UC had the smallest size at T58 for all the quarters in both the longitudinal and cross section evaluation, showing that the maximum involution of the UC size occurs at the end of a 60-day dry period. We found no statistically significant differences between the UUCS at T0 and TCPP. The amount of milk produced at the end of lactation (T0) might be similar to the colostrum production (TCPP) in terms of volume produced, thus explaining the lack of differences between these two times. The UUCSs at T0 were significantly lower than the UUCS at T7PP for the FQLS and the HQCS, while no differences were found for FQCS and HQLS. This might be due to the physiological variability in the cows’ level of milk production that could have influenced the results considering the limited number of cows included. A higher number of enrolled animals should be considered for further studies. The UCCS at TCPP was statistically lower than the UCCS at T7PP for all the quarters evaluated in both sections. This might be related to the different amount of milk produced by a dairy cow during the colostrum phase (TCPP), compared to an early lactation phase (T7PP) [30]. Ayadi and colleagues [21] reported a positive correlation between the UC size and the amount of milk produced, thus the increase in the UUCS at the same time as the increase in the milk volume produced was expected [21,30].

The bovine mammary gland can be affected by several pathological conditions; however, the most important one for both animal and human health and the economy of the dairy industry is mastitis. Studies on the diagnostic utility of mammary gland ultrasonography [20,31] have concluded that ultrasonographic examination of the intramammary structures may be useful for determining dairy cows that potentially have a higher risk of mastitis. Bovine mastitis is also related to a reduction in milk yield [34,35]; however, it is not possible to assess milk production during the dry period due to possible mastitis. Thus, using ultrasound to measure the udder cistern size during the dry period may represent an early and non-invasive screening method for diagnosing mastitis before putting the cows into the lactating group.

## 5. Conclusions

The use of UUCS during the dry period in healthy cows was easy to perform in field conditions. UUCS might give useful information for monitoring the udder during the dry period. Further studies could increase the number of animals included and the enrolled cows affected by mastitis in order to assess any possible differences between healthy and sick animals.

## Figures and Tables

**Figure 1 animals-10-02082-f001:**
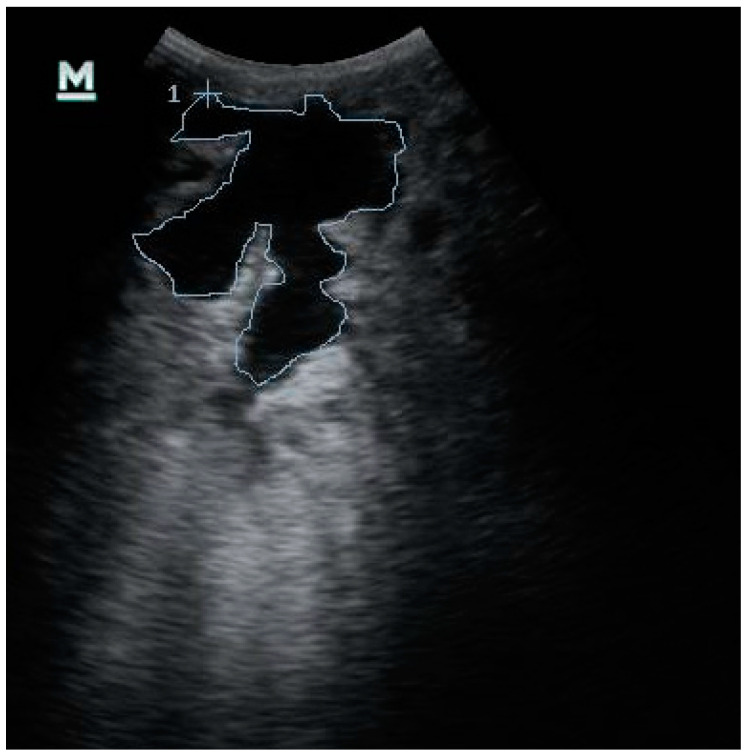
Ultrasonographic evaluation of the udder cistern size of the fore quarter in cross-section performed at T1. Freehand tracing of the udder cistern to calculate the udder cistern size using ImageJ software (National Institute of Health, Bethesda, MD, USA).

**Figure 2 animals-10-02082-f002:**
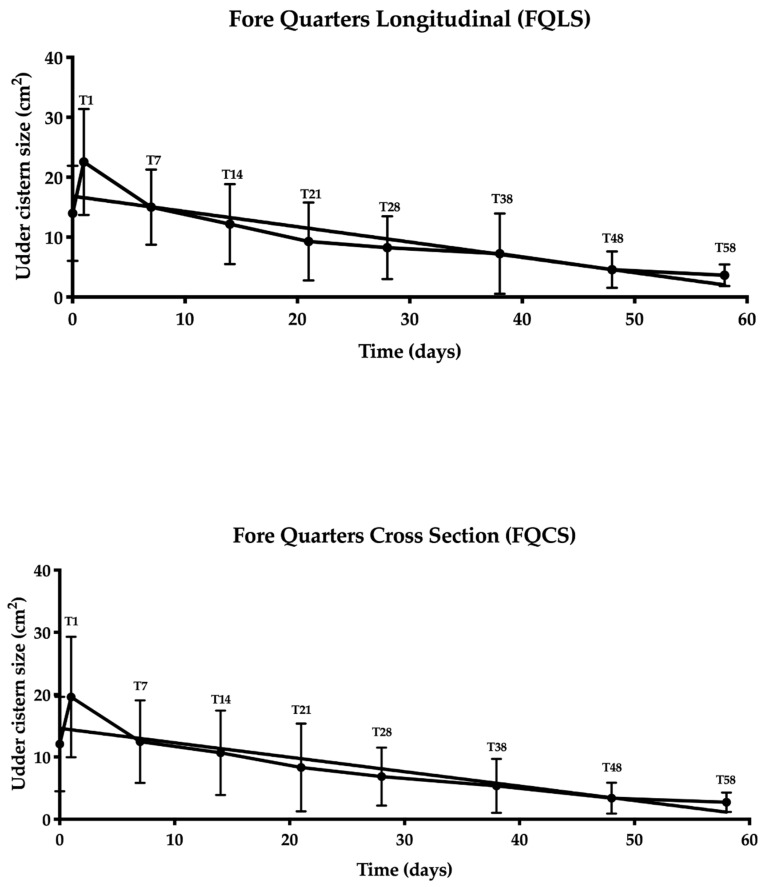
Correlation analysis between the udder cistern size and time (T0–T58) for front quarters in the longitudinal and cross sections.

**Figure 3 animals-10-02082-f003:**
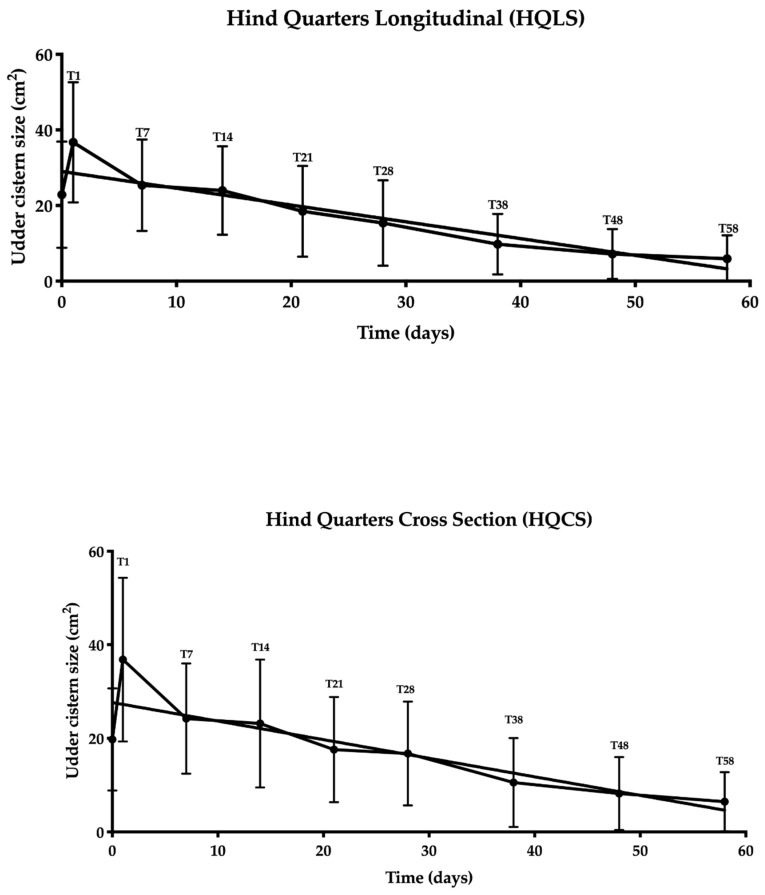
Correlation analysis between the udder cistern size and time (T0–T58) for hind quarters in the longitudinal and cross sections.

**Figure 4 animals-10-02082-f004:**
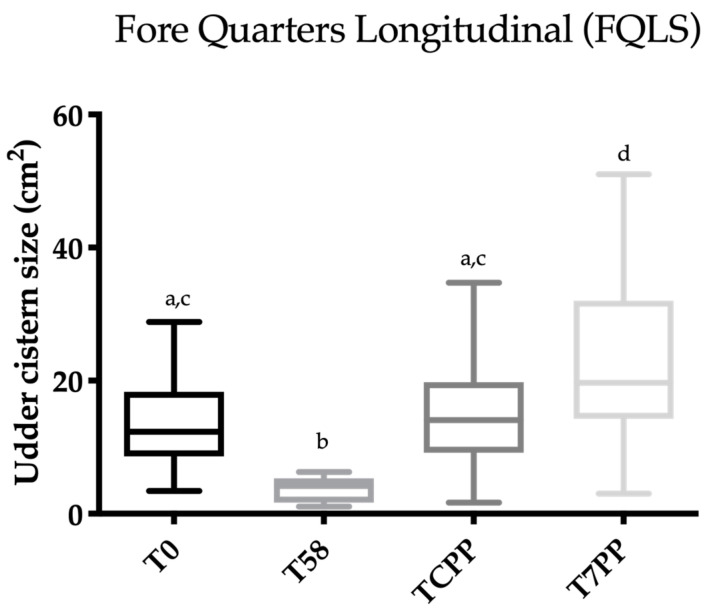
Scatter box plot showing the median, minimum and maximum values of the ultrasonographic udder cistern size between T0 vs. T58 vs. T colostrum production phase (TCPP) vs. T 7 days post-partum (T7PP) for the fore quarter longitudinal section (FQLS). Legend: a ≠ b ≠ c ≠ d: *p* < 0.05.

**Figure 5 animals-10-02082-f005:**
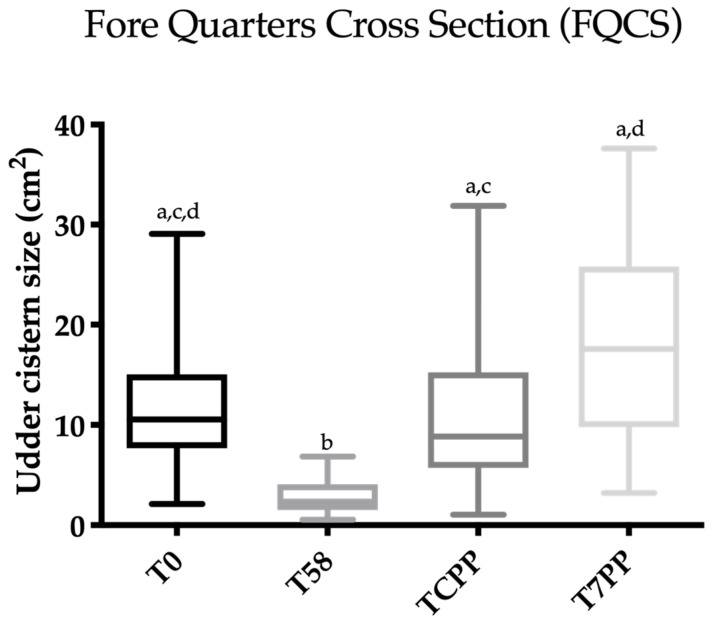
Scatter box plot showing the median, minimum and maximum values of the ultrasonographic udder cistern size between T0 vs. T58 vs. T colostrum production phase (TCPP) vs. T 7 days post-partum (T7PP) for the fore quarter cross section (FQCS). Legend: a ≠ b ≠ c ≠ d: *p* < 0.05.

**Figure 6 animals-10-02082-f006:**
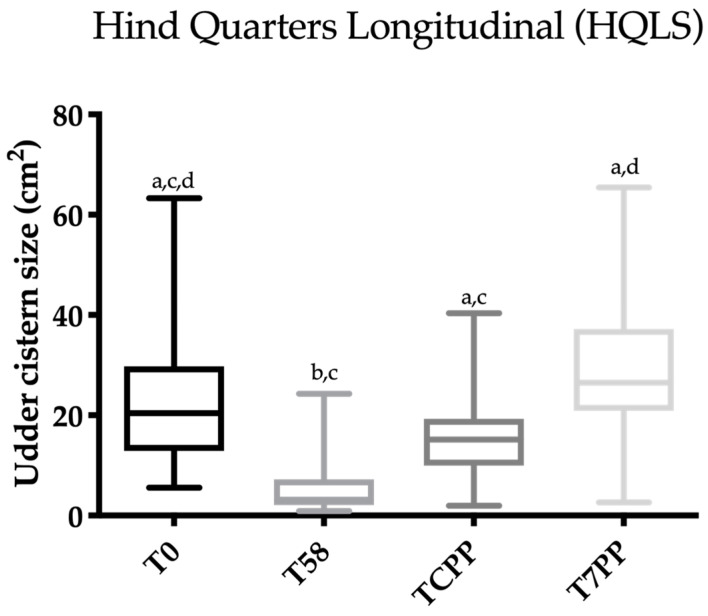
Scatter box plot showing the median, minimum and maximum values of the ultrasonographic udder cistern size between T0 vs. T58 vs. T colostrum production phase (TCPP) vs. T 7 days post-partum (T7PP) for the hind quarter longitudinal section (HQLS). Legend: a ≠ b ≠ c ≠ d: *p* < 0.05.

**Figure 7 animals-10-02082-f007:**
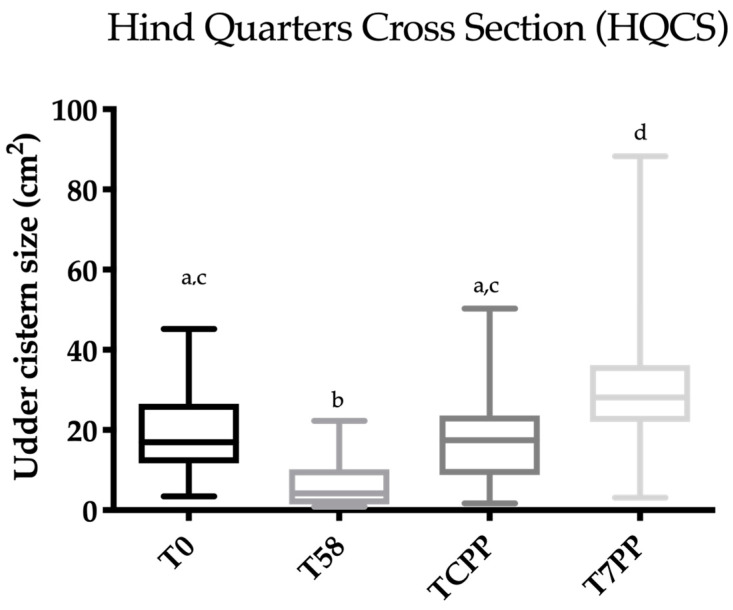
Scatter box plot showing the median, minimum and maximum values of the ultrasonographic udder cistern size between T0 vs. T58 vs. T colostrum production phase (TCPP) vs. T 7 days post-partum (T7PP) for the hind quarter cross section (HQCS). Legend: a ≠ b ≠ c ≠ d: *p* < 0.05.

**Figure 8 animals-10-02082-f008:**
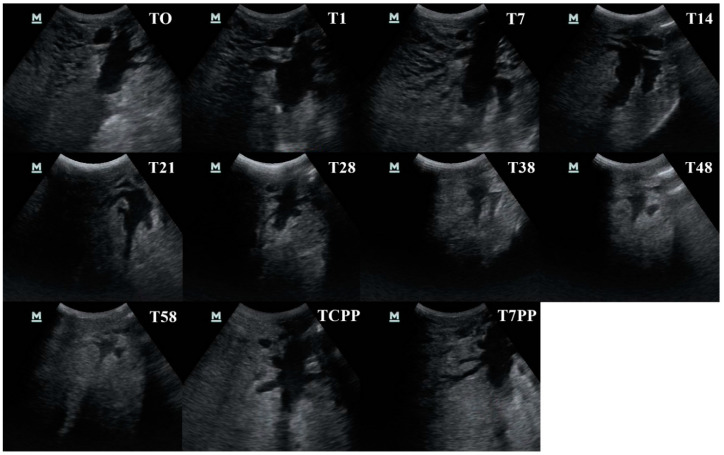
The udder cistern size as revealed by ultrasonography throughout the study period (T0, T1, T7, T14, T28, T38, T48, T58, T colostrum production phase—TCPP and T 7 days post-partum—T7PP) of the hind quarter in the longitudinal section in the same healthy cow.

**Table 1 animals-10-02082-t001:** The median, maximum (Max) and minimum (Min) values of the size (area) of the udder cistern (UC) for the front (FQ) and hind (HQ) left (-L) and right (-R) quarters in the longitudinal section for T0, T1, T7, T14, T21, T28, T38, T48, T58, T colostrum production phase (TCPP) and T 7 days post-partum (T7PP).

UC Longitudinal Section (cm^2^)		T0	T1	T7	T14	T21	T28	T38	T48	T58	TCPP	T7PP
FQLS—left	Min	3.48	10.11	7.35	5.93	2.33	1.87	1.47	1.28	1.34	1.67	3.03
	Median	14.36	24.76	15.26	13.59	7.94	9.95	5.74	3.72	3.79	13.34	21.19
	Max	28.82	35.02	24.88	18.39	20.57	15.58	16.93	9.05	5.58	34.73	51.00
FQLS—right	Min	3.43	4.12	2.96	2.11	1.06	0.72	0.83	1.02	1.08	4.15	13.20
	Median	8.94	24.75	15.20	10.56	6.30	6.35	5.30	4.06	4.28	14.09	19.05
	Max	24.65	36.48	25.22	27.26	23.88	17.70	28.95	11.16	6.28	25.54	32.21
HQLS—left	Min	7.82	19.23	10.33	14.92	6.56	4.20	0.83	0.88	0.89	4.64	11.24
	Median	26.63	37.71	22.86	20.45	14.64	10.88	7.23	3.99	4.93	16.02	26.77
	Max	63.28	63.81	54.50	53.41	49.06	40.05	27.53	20.81	24.32	40.34	65.46
HQLS—right	Min	5.58	13.57	8.34	6.49	4.35	3.12	1.21	1.37	1.42	1.94	2.61
	Median	17.98	30.53	22.57	18.95	13.68	10.29	7.78	2.92	2.80	14.31	25.23
	Max	39.15	61.56	39.96	37.16	35.35	28.89	21.71	16.16	9.32	28.57	47.89

**Table 2 animals-10-02082-t002:** The median, maximum (Max) and minimum (Min) values of the size (area) of the udder cistern (UC) for the front (FQ) and hind (HQ) left (-L) and right (-R) quarters in the cross section for T0, T1, T7, T14, T21, T28, T38, T48, T58, T colostrum production phase (TCPP) and T 7 days post-partum (T7PP).

UC Cross Section (cm^2^)		T0	T1	T7	T14	T21	T28	T38	T48	T58	TCPP	T7PP
FQLS—left	Min	2.12	6.43	7.66	2.43	1.26	1.81	0.68	0.65	0.55	1.06	3.22
	Median	12.15	17.42	12.73	11.31	8.09	6.93	4.10	2.73	2.66	8.26	17.41
	Max	26.38	35.13	21.22	19.54	17.88	14.69	11.55	8.82	6.85	26.28	37.61
FQLS—right	Min	2.31	2.80	1.25	1.28	0.68	0.41	1.62	1.21	1.22	3.31	5.33
	Median	7.82	18.32	10.08	9.21	7.95	6.69	4.11	2.94	2.42	9.27	17.75
	Max	29.08	29.00	25.21	22.02	26.52	15.11	18.84	8.91	4.18	31.87	31.88
HQLS—left	Min	9.33	17.34	9.88	9.72	4.61	3.01	1.00	0.90	0.88	1.74	3.09
	Median	18.79	33.43	21.32	15.43	14.57	10.95	6.23	6.86	6.64	18.60	28.67
	Max	45.23	88.78	49.54	51.17	35.08	34.60	29.12	24.24	22.30	50.29	88.25
HQLS—right	Min	3.45	17.33	9.79	4.97	5.06	5.22	1.00	1.20	1.08	5.61	8.02
	Median	15.74	35.40	21.22	20.74	13.13	8.73	7.70	3.23	2.82	12.61	27.82
	Max	43.04	57.41	42.83	47.35	41.42	37.00	29.24	21.36	14.14	35.58	64.74

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
