# Peer review of "Mammary Cistern Size during the Dry Period in Healthy Dairy Cows: A Preliminary Study for an Ultrasonographic Evaluation"

_animals, 2020, doi:10.3390/ani10112082_

Round 1

Reviewer 1 Report

I found the proposed research interesting and worthy of publication, even if the evidence that the evaluation of the size of the udder cistern may have during mastitis must still be considered as missing. Therefore, as reported by the Authors in the Conclusions, it is recommended to continue the research on animals with subclinical and clinical mastitis.

Could the Authors discuss any possible pathological changes related to the size of the udder cistern area in order to implement the clinical relevance of the research?

At line 95-96, you wrote " The probe was placed immediately cranial to the
insertion of each teat". Did you, probably, mean dorsal or above?

At line 113, the Authors describe how they delimited freehand the UC size. Since the research has a methodological significance, it would be useful to add an explanatory compound figure, especially in order to make the method repeatable.

At line 115, area of the UC expressed as cm3? Probably, cm2. 

As above, at Tab 1 and 2.

At line 248, better October 2018 

Reviewer 2 Report

General comment: The authors presented an interesting and original work concerning to the ultrasonographic evaluation of the mammary cistern size of healthy dairy cows during dry period.

Title: It is adequate.

Abstract: It is adequate.

The Introduction, Material and Methods, Results and Discussion are adequately described, and illustrated by several figures.

Recommendation: The manuscript should be accepted for publication.

Reviewer 3 Report

This is a very interesting paper about the ultrasonographic evaluation of the udder cistern size during the dry period and at the beginning of the subsequent lactation. As the authors said, during the dry period cows are at a high risk of developing mastitis therefore, the study of diagnostic methods useful in udder monitoring during this period are very useful. I suggest the authors to review the grammar by a native English speaker and to follow the journal's guidelines on citation style.

Abstract:

In order to make simple, clear and homogeneous the use of abbreviations in the text, I propose to abbreviate hind quarters longitudinal section with HQLS and forequarters longitudinal section with FQLS.
From line 27 to line 34 I propose to write the following:
The Friedman test and the Dunn’s test for multiple comparisons as post-hoc were performed to compare the forequarters and hindquarters cross-sections (FQCS and HQCS, respectively) and the forequarters and hindquarters longitudinal sections (FQLS and HQLS, respectively) at T0 vs T58 vs TCPP vs T7PP. A total of 440 images were evaluated. A negative linear correlation between time and FQCS and FQLS (r = -0.95; p <0.0004) and between time and HQCS and HQLS (r = -0.90; p <0.002) was found. The UUCS decreased throughout the dry period, starting to increase at the beginning of the next lactation. The measurement of the UUCS might give useful information for the dry period monitoring.

Introduction
Line 40: “underwent” should be replaced by “undergoes”

Line 44: The abbreviation “IMI” should be deleted since it does not appear in other parts of the textLine 49: “tissue” should be replaced by “tissues”

Line 49: “tissue” should be replaced by “tissues”
Line 55: “by the ultrasound technique” may be replaced by “by ultrasonography” Materials and methods
Line 66: “is” should be replaced by “was” and “are” should be replaced by “were” Line 67: “are” should be replaced with “were”
Line 71: “cover” should be replaced by “covered”
Line 72: “remove” should be replaced by “removed”

Line 73: The abbreviation “TMR” should be deleted since it does not appear in other parts of the text, “have” should be replaced by “had”

Line 74: “cover” should be replaced by “covered”
Line 78: “no heifer” may be replaced by “do not be heifer”

Lines 78-79: “no mastitis” may be replaced by “do not be affected by mastitis”
Line 80: “and four “working” quarters” may be replaced by “to have four “working” quarters”

Line 92: the sentence “The udder was never clipped or needed to be cleaned before the exam.” May be replaced by “The udders were not clipped or cleaned before the ultrasonographic examination”.

Line 96: the sentence “to the insertion of each teat on the udder” may be replaced by “to the teat-udder insertion”

Line 97: please specify which UC size was calculated (I suppose area) Line 99: “store” should be replaced by “stored”

Line 100: “on a week base” should be replaced by “once a week”
Line 103: “ultrasound evaluation” may be replaced by “ultrasonographic evaluation” Line 104: “no” should be replaced by “not”. I suggest deleting “milk the cow and”

Lines 107-108: I suggest deleting the sentence “Subjects with signs of subclinical or clinical mastitis were excluded from the study.” because authors have already specified it in line 79.

Lines 110-111: please move the sentence “The pictures were evaluated by the software ImageJ (National Institute of Health, USA).” at the beginning of the paragraph 2.4. Ultrasonographic images evaluation

Line 115: the area should be expressed in cm2 not in cm3 Line 117: “images” should be replaced by “image”.

Lines 119-120: authors should write “data were expressed as median and rage or as mean and standard deviation (SD)” in fact, in the Results section, you report age, body weight, BCS, SCC at T0 and T7PP as mean and SD, so I suppose that all this variables are normally distributed. You should delate “standard error” because it is not a measure of central tendency

Line 120-125: In order to simplify the use of abbreviation you may write: “Data collected were analyzed with the Spearman test for verifying the correlation between UCCS and time (T0-T58). In order to compare the FQCS, HQCS, FQLS and HQLS at T0 vs T58 vs TCPP vs T7PP a Friedman test with a Dunn’s test for multiple comparison were used. “

Results

Line 131: please add “±SD” after “mean” and report the standard deviation of the body weight and of BCS Lines 134-135: please add “±SD” after “mean”

Table 1: In order to simplify and make homogeneous the use of abbreviation you may replace FQ-L with FQLS-left, FQ-R with FQLS-right, HQ-L with HQLS-left and HQ-R with HQLS-right

Table 2: In order to simplify and make homogeneous the use of abbreviation you may replace FQ-L with FQLS-left, FQ-R with FQLS-right, HQ-L with HQLS-left and HQ-R with HQLS-right

Line 143: I suppose that “the size of the udder cistern” is the area measured in longitudinal section, if so, please specify it and report cm2 instead of cm3

Line 147: I suppose that “the size of the udder cistern” is the area measured in cross section, if so, please specify it and report cm2 instead of cm3

Line 152: please replace FQL with FQLS and HQL with HQCS Graph 1: please replace FQL with FQLS and FCR with FQCS

Line 155: please replace “ultrasonographic ultrasound cistern size” with “ultrasonographic udder cistern size” and delete the abbreviation UUCS because it does not appear in the graph

Graph 2: please replace HQL with HQLS
Line 158: please replace “ultrasonographic ultrasound cistern size” with “ultrasonographic udder cistern

size” and delete the abbreviation UUCS because it does not appear in the graph Lines 161-166: please replace FQL with FQLS and HQL with HQCS
Graph 3: please replace FQL with FQLS

Lines 168-170: please replace “ultrasonographic ultrasound cistern size” with “ultrasonographic udder cistern size”, delete the abbreviation UUCS because it does not appear in the graph and replace FQL with FQLS

Lines 172-174: please replace “ultrasonographic ultrasound cistern size” with “ultrasonographic udder cistern size” and delete the abbreviation UUCS because it does not appear in the graph

Graph 5: please replace HQL with HQLS
Lines 176-178: please replace “ultrasonographic ultrasound cistern size” with “ultrasonographic udder

cistern size”, delete the abbreviation UUCS and replace HQL with HQLS

Lines 180-182 please replace “ultrasonographic ultrasound cistern size” with “ultrasonographic udder cistern size” and delete the abbreviation UUCS

Line 191: please replace “ultrasonographic ultrasound cistern size” with “ultrasonographic udder cistern size”

Discussion
Line 196: “tissue” should be replaced by “tissues”
Line 215: “differed” should be replaced by “differ”
Line 226: “statistically” should be replaced by “significantly” Line 226: please replace FQL with FQLS
Line 227: please replace HQL with HQLS
